# Use of Virtual Reality and Videogames in the Physiotherapy Treatment of Stroke Patients: A Pilot Randomized Controlled Trial

**DOI:** 10.3390/ijerph20064747

**Published:** 2023-03-08

**Authors:** Francisco-Javier Peláez-Vélez, Martina Eckert, Mariano Gacto-Sánchez, Ángel Martínez-Carrasco

**Affiliations:** 1Doctoral Program in Health, Disability, Dependency and Wellness, Campus de Ciencias de la Salud, University of Murcia, 30120 Murcia, Spain; 2Group on Acoustics and MultiMedia Applications (GAMMA), Centro de Investigación en Tecnologías Software y Sistemas Multimedia Para la Sostenibilidad (CITSEM), Universidad Politécnica de Madrid (UPM), 28031 Madrid, Spain; 3Department of Physiotherapy, Campus de Ciencias de la Salud, University of Murcia, 30120 Murcia, Spain

**Keywords:** physiotherapy techniques, cerebrovascular stroke, virtual reality, rehabilitation, exergaming

## Abstract

A stroke is a neurological condition with a high impact in terms of physical disability in the adult population, requiring specific and effective rehabilitative approaches. Virtual reality (VR), a technological approach in constant evolution, has great applicability in many fields of rehabilitation, including strokes. The aim of this study was to analyze the effects of a traditional neurological physiotherapy-based approach combined with the implementation of a specific VR-based program in the treatment of patients following rehabilitation after a stroke. Participants (*n* = 24) diagnosed with a stroke in the last six months were randomly allocated into a control group (*n* = 12) and an experimental group (*n* = 12). Both groups received one-hour sessions of neurological physiotherapy over 6 weeks, whilst the experimental group was, in addition, supplemented with VR. Patients were assessed through the Daniels and Worthingham Scale, Modified Ashworth Scale, Motor Index, Trunk Control Test, Tinetti Balance Scale, Berg Balance Scale and the Functional Ambulation Classification of the Hospital of Sagunto. Statistically significant improvements were obtained in the experimental group with respect to the control group on the Motricity Index (*p* = 0.005), Trunk Control Test (*p* = 0.008), Tinetti Balance Scale (*p* = 0.004), Berg Balance Scale (*p* = 0.007) and the Functional Ambulation Classification of the Hospital of Sagunto (*p* = 0.038). The use of VR in addition to the traditional physiotherapy approach is a useful strategy in the treatment of strokes.

## 1. Introduction

A stroke, also known as a cerebrovascular accident (CVA), is one of the main causes of physical disability in adults [1]. It is a neurological disease characterized by the dysfunction of different areas of the brain caused by a sudden vascular (either arterial or venous) disease [2,3].

A stroke can be either ischemic or hemorrhagic. An ischemic stroke, which is the most frequent type, is caused by a significant decrease in the blood flow to a part of the brain. This lack of blood supply produces a cerebral infarction, generated by an embolus, leading to the death of neurons due to a lack of oxygen supply and nutrients in the blood [2]. On the other hand, hemorrhagic strokes are associated with higher mortality rates, however, in the long term, recovery from the sequelae is usually better. They are the result of a hemorrhage caused by the rupture of a cerebral vessel, causing bleeding and extravasation into the cerebral space [2].

Strokes are currently the second leading cause of death and the leading cause of disability in Europe [3,4]. In Spain, they are the second leading cause of death, the leading cause of disability and the second leading cause of dementia [5]. Every 6 min, a new case of stroke occurs in Spain [6]. Subjects who have experienced a stroke-based episode suffer from a wide and variable combination of physical, cognitive, emotional and behavioral problems and functional difficulties in the activities of daily living that lead to a decrease in the degree of participation, integration and quality of life [7]. Arterial hypertension is the most frequent modifiable risk factor and the one that contributes most to the onset of this pathology [8]. The most common physical sequelae of strokes are usually increased spasticity [9], sudden decrease in strength [10,11], loss of balance and postural control of the trunk and decreased functional gait ability [12,13,14].

Neurorehabilitation is a therapeutical approach which mainly focuses on neurological care and the early application of rehabilitative treatment, with a wide range of therapeutic applications [15]. It is a complex care process aimed at restoring, minimizing or compensating for (to the highest possible extent) any functional deficit derived from a central nervous system injury [16]. The rehabilitation of patients after a stroke has become a highly complex care process with high personal, family and social costs, and it requires the existence of highly trained multidisciplinary teams for its proper management: the combination of conventional rehabilitation techniques alongside the introduction of new technologies supported by the most updated scientific evidence becomes, therefore, a cornerstone in the treatment of strokes [17,18].

Virtual reality (VR) is the creation of an artificial scenario adapted to the real world, in which it is possible to interact, navigate and immerse in a three-dimensional space by applying different sensory stimuli. Thus, it implies a full-immersion process of the person or the patient in a virtual scenario as close as possible to the real world [19]. Its beginnings go back to the first quarter of the 20th century [20]. More recently, in the 21st century, much progress has been made in the technological field, resulting in the creation of different devices and VR systems [21,22]. In non-immersive VR, there is full awareness of the outside world [23]; semi-immersive VR is the combination of VR technology and simulator-based technology [24], with retained awareness of the outside world [25]. Immersive VR is the generation of an environment in three dimensions (3D), where the subject is completely immersed in the experience by means of “Head-Mounted Display” (HMD) devices (e.g., VR glasses). These devices have a screen for each eye and allow the recreation of a virtual world in 3D in which the user can play while being completely immersed in the experience without having contact with the outside world [26].

VR can be used in the clinical field as a non-invasive technological therapy that allows the patient to interact with a computer-created environment. The use of VR and video games in the health sector increasingly requires more evidence of their efficacy in combination with conventional treatments. This efficacy is explained by their numerous applications and beneficial effects in patients affected by neurological, psychiatric or musculoskeletal pathologies [27]. The implementation of this technology in the field of neurology is of paramount importance, since there is a large number of studies supporting and endorsing the beneficial effects on the patient, specifically in stroke rehabilitation [28,29], providing improvements in gait, postural balance, trunk balance, strength, spasticity, upper limb motor function and improvements in cognitive level and attention, among others [30,31,32,33,34,35,36,37,38,39].

One of the physiological explanations for these physical improvements is the action of mirror neurons. According to Rizzolatti (discoverer of mirror neurons), the essence of this “mirror” mechanism is the following: each time individuals observe an action performed by another person, a set of neurons that encode that action is activated in the motor system of the observers. If the patient observes a certain action in the VR environment, they will be “activated” at a motor level by the functioning of their mirror neurons [40,41].

The aim of the current study is to analyze the effects of a traditional neurological physiotherapy-based approach combined with the implementation of a specific VR-based program in the treatment of patients following rehabilitation after a stroke.

## 2. Materials and Methods

A description of the methods followed for the recruitment of participants, the design of the interventions, the process of data collection and the overview of the statistical analyses are provided in the current section.

### 2.1. Participants

The current study corresponds to a pilot randomized clinical trial involving 26 patients from the outpatient clinical setting “Casaverde” in Murcia (Spain).

The inclusion criteria were patients receiving neurological physical therapy treatments, patients having suffered a stroke (hemorrhagic or ischemic) in the last 6 months, an 18- to 80-year age range, ability to have active mobility in at least one upper limb, ability to understand simple instructions and ability to keep a standing position. The exclusion criteria concerned patients with neurodegenerative diseases such as Alzheimer’s and Parkinson’s, having undergone surgery on the limbs assessed or having scars which limit mobility.

The corresponding informed consent, signed by all the participants or their legal caregivers, was collected prior to the onset of the study.

Participants who met the inclusion criteria received a code and were subsequently included in one of the two groups. Assignment to the groups was randomized by numerical codes using the “Random Group Generator” tool of “RecursosTic.net”. The physical therapist who applied the VR treatment was not blinded to the use of both VR and neurological physical therapy treatment in each patient. The blinding itself corresponded to the assessors, since they were not aware of the assignment of any of the patients evaluated, neither at the baseline nor at the post-test measurement (after six weeks).

A total of 26 subjects were initially recruited. Two subjects dropped out of the study for the following reasons: (i) considering the videogame and VR approach a waste of time and (ii) nonattendance to the clinical setting for 2 weeks. Therefore, 24 patients participated in the study and were randomly assigned to either a control group (CG, 12 subjects) or an experimental group (EG, 12 subjects). Figure 1 displays the flowchart of the study.

The study adhered to the Declaration of Helsinki, it was approved by the Ethics Committee of the University of Murcia under the code JVP-2022-01 and it was registered in the U.S. National Library of Medicine Clinical Trials.Gov with the identifier NCT05278403.

### 2.2. Intervention

This study lasted 6 weeks. Both groups received a 1-h neurological physiotherapy session 5 times a week, but the EG additionally received 3 sessions of VR per week. This pattern of 3 VR sessions per week was previously described and used by Cho et al. [36] and Abd El-Kafy et al. [37].

The neurological physiotherapy sessions consisted of upper and lower limb strengthening exercises, resistance exercises, active and active-assisted kinesitherapy, anti-resistance exercises, aerobic exercises, simple and obstacle parallel bar gait, jumping gait, skill circuits with different tasks and different surfaces, stable and unstable, work on trellises, fine motor skill exercises in both limbs and weight transfer exercises to improve balance and postural stability. The exercises were performed in different positions, including standing, sedentary, quadrupedia, supine and prone depending on the condition of each patient.

The VR program applied was an immersive VR by means of VR glasses (Oculus Quest 2), a computer (with the software), a camera (Kinect 360 v1) used as a sensor of the patient’s movements and a router that acted as a connection point between the glasses and the computer (Figure 2). In addition, an application was used to reproduce the image of the computer software in the VR glasses (Virtual Desktop), since it was possible to send this image to the glasses through the router and create the virtual environment. While the patient performed the exercises visualized in the VR glasses, the physiotherapist, located behind the patient, supervised the development of the video game thanks to the computer screen, therefore monitoring both the patient and the video game.

For the generation of the virtual environment, a specific software called “Blexer “ was used, which was created with the idea of being a rehabilitation tool for physiotherapists [42,43,44]. Within this software, a game called “Phiby’s Adventure” was used, which is a video game composed of four “mini games”. All patients performed the VR sessions in a standing position. Initially, a tailored calibration adapted to each patient was performed and was subsequently used across the different sessions. Figure 3 shows a patient during a VR session, whereas Figure 4 displays two different screens of the game.

“Phiby’s Adventure” has two different modes according to each patient’s laterality. The patient, thus, by means of their left or right arm, will have to direct the green character, called “Phiby”, to one of the following mini games: Chop the Wood (chopping movement), Row the Boat (performing rowing movements) or Climb the Tree (simulating the movement of climbing up a tree) (Figure 5). Although four mini games are available, solely the aforementioned three were used due to the difficulty of the fourth mini game.

To complete all the mini games, patients had to achieve a target of logs cut, meters rowed or meters climbed within a time limit. An interesting aspect of this software is that both the objectives and the time limit can be set according to the criteria of the physiotherapist prior to the exercise, depending on the ability of each patient, so that it is not the patient who adapts to the game: it is the game that adapts to each patient’s situation.

Initially, a first contact session is carried out, with a time limit set at 180 s and a target of 40 logs cut, 40 m rowed and 40 m climbed, so that the patient has time to delve into the environment and perform the test.

Both the calibration and the use of the mini games were performed in a standing position, with the physiotherapist behind the patient giving positive reinforcement and assisting the patient, if necessary, in each mini game. A chair was placed next to the patient as a safety measure, in case the patient suffers a headache, dizziness or needs to rest.

The VR therapy was always applied after the neurological physiotherapy treatment.

### 2.3. Data Collection

All researchers involved in data collection were trained in and practiced the standardized data collection and measurement procedures.

Initially, sociodemographic data (gender and age) were collected.

Measurements of the different clinical variables were taken at baseline (pre-treatment or baseline) and at the end of treatment (post-treatment) after the corresponding six weeks of treatment. The different clinical variables were:Muscle strength, measured through the Daniels and Worthingham Scale. The following muscles were assessed: shoulder flexors, elbow flexors, wrist flexors, hip flexors, knee flexors, ankle flexors, shoulder extensors, elbow extensors, wrist extensors, hip extensors, knee extensors and ankle extensors.Spasticity, measured by means of the Modified Ashworth Scale on shoulder flexors, elbow flexors, wrist flexors, hip and knee flexors, ankle and foot flexors, shoulder extensors, elbow extensors, wrist extensors, hip and knee extensors, ankle and foot extensors.Functionality was assessed through the Motricity Index. Arms and legs were tested: pinch grip, elbow flexion, shoulder abduction, ankle dorsiflexion, knee extension and hip flexion.Trunk control was assessed through the following movements: rolling to weak side, rolling to strong side, sitting up from lying down and balance in the sitting position.Balance and gait were evaluated through the Tinetti Balance Scale.Balance was assessed by means of the Berg Balance Scale.The Functional Ambulation Classification of the Hospital of Sagunto (CFMHS) was used for the assessment of the functional level of gait.

### 2.4. Statistical Analyses

Means between sex and age groups were analyzed and significant differences were sought between the groups. Data were assessed for normality through the Shapiro–Wilk test due to the small sample size (*n* = 24). Intra-group comparisons between the baseline and post-test scores across the different variables were assessed by means of *t*-tests or Mann–Whitney U tests, when applicable, for continuous variables, whereas comparisons for categorical variables were carried out using chi-squared tests (or a chi-squared test with Yates correction whenever more than 20% of the frequencies were “zero”).

In relation to the required sample size, a calculation using the G* Power software (ver. 3.1.9.7; Heinrich-Heine-Universität Düsseldorf, Düsseldorf, Germany) was performed using the following parameters: one tail, medium effect size d = 0.3, alpha = 0.05, statistical power = 0.80 and a 1:1 allocation ratio. The total sample size would consist of 64 subjects (i.e., 32 individuals per group); our study therefore corresponds to a pilot study, since a total of 24 individuals completed the study (EG = 12; SG = 12). The perception that pilot trials are simply a casual prelude to a larger trial may somehow threaten the rigor with which they are implemented, and such an approach to pilot trial design and implementation runs the risk of providing misleading results, as stated by Arnold et al. [45]. For this reason, the current study followed the recommendations for reporting the results of pilot studies developed by Thabane et al. [46].

All analyses were performed using the statistical package SPSS v.24 (Statistical Package for the Social Sciences, IBM Corp, Armonk, NY, USA). An alpha value of 0.05 was set across all the analyses performed.

## 3. Results

This study involved 24 stroke patients. The CG had seven males and five females, with a mean age of 59.58 ± 15.97 years (Median: 65 years; age range: 18 to 75 years of age). The EG encompassed nine males and three females, with a mean age of 51.91 ± 18.58 (Median: 58 years; age range: 18 to 71 years of age). No statistically significant differences were found between the two groups, neither for gender distribution (*p* = 0.667) nor for age (*p* = 0.293). Of the 24 participants, 13 had a right-sided condition and 11 had a left-sided condition as a result of a stroke. All the strokes had occurred in the last 6 months, and all patients were under long-term recovery in their own homes (not institutionalized) and attending outpatient rehabilitation, according to the stages reported by Watson and Quinn [47].

Concerning the post-test assessment (after six weeks of treatment), muscle strength revealed no differences between both groups for any of the muscular groups assessed, with p-values ranging from 0.154 (wrist-extensors) to 0.897 (elbow-flexors). Likewise, spasticity did not experience significantly different changes between both groups across the different muscle groups assessed, with a *p*-value range from 0.227 (hip-flexors) to 0.882 (knee-flexors).

Data on functionality, assessed through the Motricity Index, revealed the existence of statistically significant differences in the Trunk Control Test, Tinetti, Berg Scale and the CFMHS variables in the EG, but not in the CG. Data on the aforementioned results are displayed in Table 1.

## 4. Discussion

According to the results obtained, the use of neurological physiotherapy in combination with VR and video games is more effective for the treatment of patients following a stroke than neurological physiotherapy alone. This 6-week pilot randomized clinical trialhas shown significant improvements in the Motricity Index, Trunk Control Test, Tinetti Balance Scale, Berg Balance Scale and Functional Ambulation Classification of the Hospital of Sagunto. Thus, the experimental therapeutic approach used in the rehabilitation of stroke patients is an effective and efficient tool for improving the values of balance, gait, functional level of gait, trunk control and reduction of motor deficit. With regards to the evidence collected on strength (Daniels and Worthingham Scale) and spasticity (Modified Ashworth Scale), no significant improvements were found in any of the groups.

Compared to the results of other studies [1,48], our results support and endorse the efficacy of this therapeutic approach in improving balance and gait ability in people following a stroke. Concerning balance, some studies [1,36] assessed static balance by means of the Berg scale, similarly to this study. However, the dynamic analysis of balance and gait is evaluated by means of the Tinetti scale in our study, whereas other authors focused on the Timed Up & Go (TUG) and the 10 Meters Walk Test (10 MWT) [36,48,49], therefore hindering any potential comparison. A study by Cho et al. [36] used a treadmill in the control group and a treadmill and an immersive VR system in the experimental group, with a sample similar to ours in terms of sample size and mean age (CG: 63.5 ± 5.5; GI: 65.8 ± 5.7). Thestudy found statistically significant differences in both groups in dynamic balance and gait, except for static balance, and a statistically significant improvement in both groups after the intervention in dynamic balance and gait, with no significant correlation between postural sway and other dependent variables. Considering the sample size, the frequency (3 times per week) and the length of the intervention (6 weeks), our results share homogeneity to a high extent. Different articles use VR glasses [31,36,50,51] with significant differences in balance and gait between the intervention and control groups. In the study by Lee et al. [50], in which a game based on canoe movement is used, similar to the one in our study, significant results were found in postural balance and upper extremity motor function. The systematic review by Rutkowski et al. [28] analyzed twenty articles based on the use of VR in different neurological and orthopedic pathologies, highlighting that the use of specialized VR approaches can improve balance in neurological patients.

To understand the incidence of our study at a brain-processing level, some authors [52] explored brain activity by means of encephalograms and assessed the performance of VR on motor improvement in subjects with chronic hemiparesis, a complication that many stroke patients experience. This study suggests that the use of VR in combination with cycloergometer exercises provides an improvement in gait and dynamic balance Tinetti subdimensions. Furthermore, after analyzing the encephalograms, it can be seen that this type of treatment acts on different brain areas, probably at a mirror neuron system [53] involved in task planning, with results showing improvement at the motor level. Regarding spasticity, unlike in our study, other studies [37,54] show evidence of the decrease in spasticity with the use of VR and video games in stroke patients.

Focusing on strength, unlike in our study, some authors [44] disclose that, with the use of VR in combination with conventional physical exercise treatment, significant differences in strength levels are found, and that the inclusion of this therapy is highly recommended in post-stroke patients. The differences in the strength and spasticity results of these studies with respect to our study may be due to several factors. One of them may be the frequency and duration of treatment. Whilst our study presents a frequency of five days per week, with one hour per day of VR-based treatment, several authors developed longer treatment lengths compared with our study, with up to 12 weeks of total treatment periods [37].

Another important factor to highlight in the comparison with some of the studies previously developed [51,54,55] is the difference in the VR instrument used; our study focuses on the use of VR goggles, generating a completely immersive virtual environment for the patient. This environment has the limitation that the subjects to whom it is applied have a treatment time limit, since the prolonged use of these types of glasses (more than 20 min) can cause dizziness and headaches in the subjects. Therefore, in our therapy, we set the maximum time at 15 min as a safety measure for the patients. Some studies, which did not use immersive VR, may therefore have had longer treatment periods [31,32,33,34,35,51].

Our study should be interpreted in the light of its potential methodological limitations. First, a possible limitation of this study may be that the sample is not very large, although the review of other studies [32,33,34,35,48,56] shows similar sample sizes. By having a larger sample, the results could be considered more representative. Further studies with larger sample sizes are therefore required to evaluate the efficacy of these VR intervention programs. Second, the possible previous treatments undergone by the subjects within the context of their stroke were not considered or assessed. The study was carefully designed so that all patients were in similar stages of recovery, and baseline measures were compared to ensure that the initial status was taken into account. Further studies could nonetheless consider previous therapeutical approaches and/or time after event related variables, if only for descriptive purposes. Third, even though the current evidence seems to support the implementation of the use of VR and video games in the rehabilitation of stroke patients, the “enactment” of the most effective VR-based modality (i.e., real-world games [36], use of assisted robotics [37]) remains unexplored, and further research in the field, focusing on the comparison of different VR-based approaches, as well as the use of participatory designs involving different strata of the therapeutic scene [39], is required in the future.

## 5. Conclusions and Future Work

The use of VR alongside a traditional physiotherapy approach in the treatment of patients following a stroke provided significant improvements in terms of balance, gait, trunk control and functional level of gait, while no significant differences were found in strength or spasticity. The results stemming from our study highlight the efficacy of the use of VR and video games in the rehabilitation of stroke patients.

The treatment of neurological physiotherapy in combination with the use of VR and video games in the rehabilitation of stroke patients is a useful strategy. Further research should focus on larger samples, alongside establishing the most effective VR strategy and tailored interventions based on the individuals’ specific needs and deficits. Virtual reality emerges as an efficient therapeutic approach for the treatment of post-stroke-related disability, and future research may explore cheaper, safer and more accessible devices that will enable and enhance at-home rehabilitation.

## 6. Patents

The game and the Blexer environment led to several registrations of intellectual property for software with the following entry numbers: Blexer-med web platform 16/2019/1687; Middleware Chiro 16/2019/1576; and Phiby’s Adventures 16/2019/871.

The Blexer environment was loaned by the Universidad Politécnica de Madrid to the Universidad de Murcia for research and teaching purposes.

## Figures and Tables

**Figure 1 ijerph-20-04747-f001:**
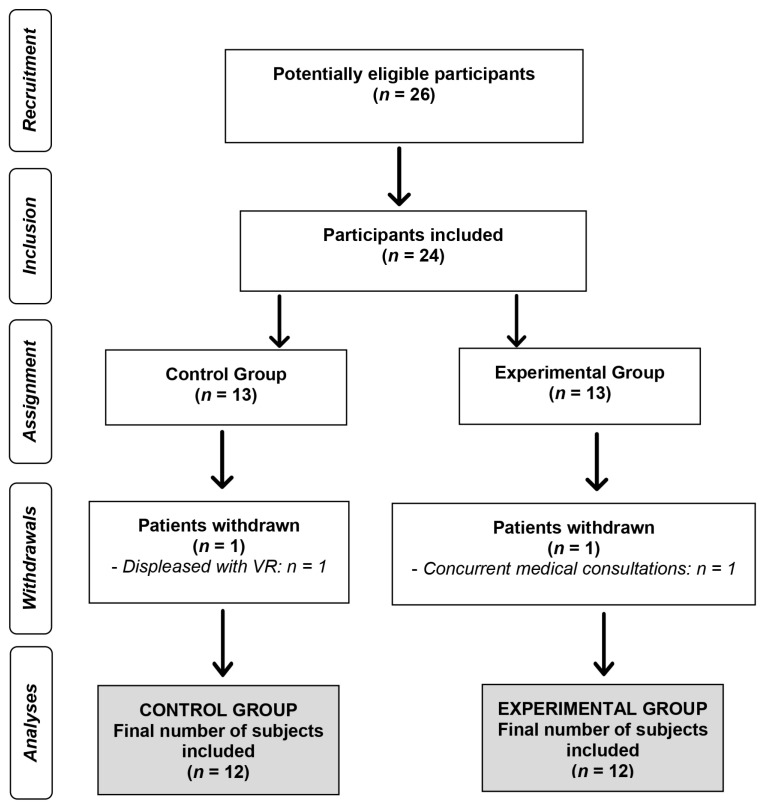
Flowchart of the study.

**Figure 2 ijerph-20-04747-f002:**
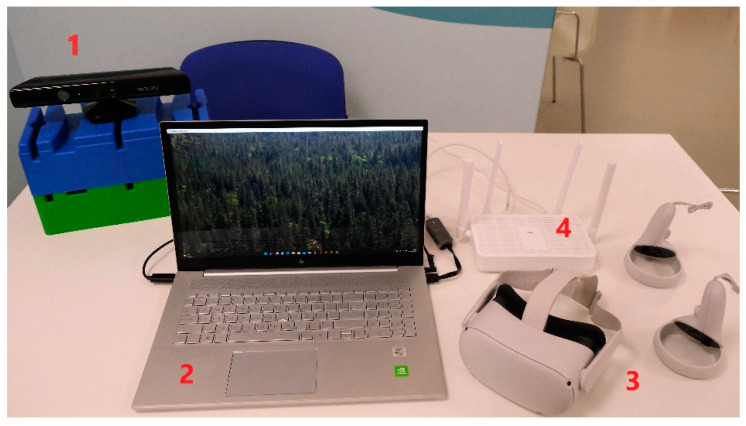
VR equipment used. 1. Kinect 360 2. Computer with the software. 3. VR glasses (oculus Quest 2) 4. Router.

**Figure 3 ijerph-20-04747-f003:**
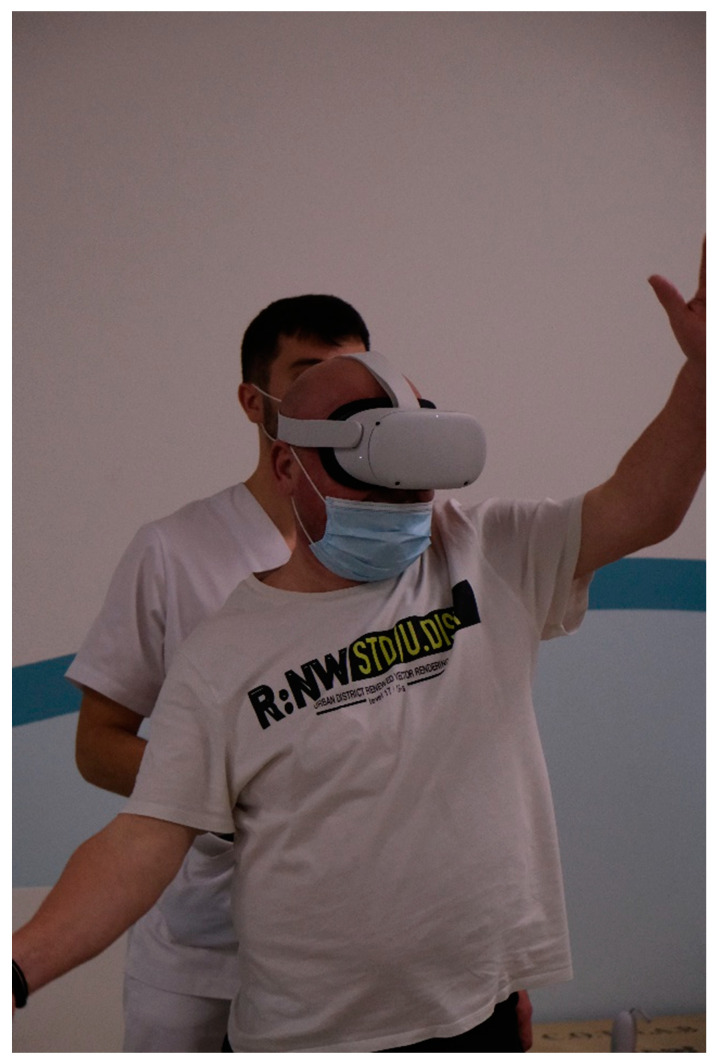
Patient during a VR session.

**Figure 4 ijerph-20-04747-f004:**
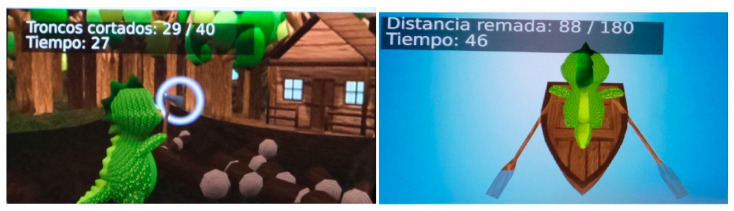
Two different screens of Phiby’s Adventure.

**Figure 5 ijerph-20-04747-f005:**
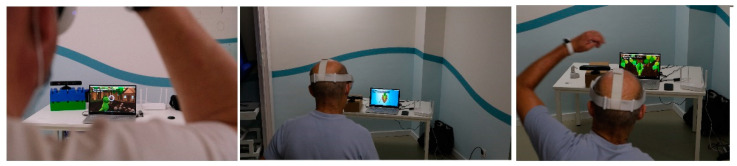
Patient playing “Chop the Wood”, “Row the Boat” and “Climb the Tree”, respectively.

**Table 1 ijerph-20-04747-t001:** Results of Clinical Variables (Mean ± SD).

Clinical Variables		CG			EG		
		Pre	Post	*p*-Value	Pre	Post	*p*-Value
Functionality (Motricity Index)		72.70 ± 37.74	75.66 ± 36.72	0.754	67.08 ± 31.66	84.00 ± 23.05	0.001
Trunk Control		46.58 ± 33.58	69.65 ± 29.39	0.083	65.16 ± 23.33	91.58 ± 21.55	0.008
Balance (Tinetti)		6.50 ± 6.08	9.08 ± 5.66	0.251	8.58 ± 4.12	13.58 ± 3.05	0.004
Gait (Tinetti)		3.58 ± 4.52	5.83 ± 3.99	0.105	5.41 ± 3.02	9.16 ± 2.85	0.006
Balance (Berg)		21.33 ± 22.76	28.91 ± 20.40	0.111	27.00 ± 15.89	46.00 ± 13.08	0.007
Functional Level of Gait (CFMHS)	0	5	0	0.280	1	0	0.038
	1	2	7		7	1	
	2	4	1		2	1	
	3	1	2		1	2	
	4	0	2		1	6	
	5	0	0		0	2	

CG: Control Group; EG: Experimental Group; Pre: Baseline measurement; Post: Follow-up measurement.

## Data Availability

The data presented in this study are available on request from the corresponding author. The data are not publicly available due to privacy issues.

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
