# Peer review of "Use of Virtual Reality and Videogames in the Physiotherapy Treatment of Stroke Patients: A Pilot Randomized Controlled Trial"

_ijerph, 2023, doi:10.3390/ijerph20064747_

Round 1

Reviewer 1 Report

I would like to begin by greeting the authors and congratulating them for having decided to investigate an area where there is still so much to discover, but also for having decided to share this article with the rest of the scientific community, so that science can evolve.

This is a Pilot Randomized-Controlled Trial on the Use of Virtual Reality and Videogames in the Physiotherapy Treatment of Stroke Patients.

All comments, questions and suggestions made are in the constructive sense and try to improve the article, after several attentive readings.

Title

Appropriate, concise and well-explained title.

Abstract

Well structured according to the rules of the journal.

Key words

Repetitions with expressions that are in the title should be avoided. Whenever possible keywords should be Mesh.

Introduction

The purpose indicated at the end of the Introduction must be exactly the same as the Abstract.

The introduction contains 37 bibliographical references. But none of them are referenced in the article.

Materials and methods

“The patients did not know 108 to which group they belonged”. I would like clarification on this statement. If the patients had their VR glasses on, how did they not know they belonged to this group?

Figure 1 should have more quality.

According to ClinicalTrials.gov, it is described that 21 participants with recruitment completed status participate in this trial. In the article authors refer to 24. I would like clarification on this.

Were these patients undergoing physical therapy prior to the start of the study? If so, how did the researchers validate this bias?

I wish the photos were better quality. It greatly improved the quality of the article and did justice to the quality of the work presented.

Results

Tables must be formatted according to the journal rules. Acronyms must be described.

More descriptive age data would be interesting, such as minimum and maximum ages for each group. In addition, the time of institutionalization and the stroke event should be described in a table, in order to try to understand the difference, or not, in the results presented.

Discussion

I enjoyed reading your discussion.

The limitations presented are honest and seem clear to me. But they may not be the only ones, depending on the authors' response to the question above.

Conclusions

I would have liked to have read a conclusion more directed towards clinical practice and how this study may or may not help to change the implemented practices. What does this study contribute to the future? I would like to see, at the end of the discussion, more directions for the future. Is virtual reality really here to stay? How can it be implemented and made available globally to the population? Will it serve all patients who have suffered a stroke?

General Comments

Acceptable English level.

It is a very interesting, innovative and very interesting topic. Overall, the authors are to be congratulated.

The article deserves some changes

Author Response

Please see the attachment. Thank you very much.

Yours,

Mariano Gacto

Reviewer 2 Report

Overall the work is well-written and organized. 

The topic is relevant and may capture future readers' attention moving forward.

Still, some comments/suggestions are described next:

Abstract

- Try to add 2/3 more lines explaining the problem, to better illustrate what is the challenge being addressed.

- Add rehabilitation to the keywords.

Introduction

- Introduce the acronym VR in the first appearance - line 55

- Then, down the paper, use the acronym instead of the term.

- The introduction could benefit from more detail on the use of video games and VR for the topic at hand. Some examples of previous work from other teams should be provided.

- A paragraph with the paper structure should be added.

Materials and methods

- An initial paragraph should be added to explain what is going to be addressed in this section.

- What was the rationale behind: "3 sessions of VR per week" ? Did a therapist or other healthcare member propose this approach? If so, justify it. 

- Please provide more images of the mini-games used, without making readers look at other publications.

- Figure 3 is blurred, and cannot be read - please fix this issue.

- What was the difficulty of the fourth mini-game? was this understood before or during the study?

- The mini-games seem to combine distinct body movements, can this be applied to all patients? Was this considered a priori when recruiting the participants?

- http://www.gpower.hhu.de/ - should be a fotnote

Results

- It appears that Muscle Strength and Spasticity were not mentioned. Can you clarify?

- What was the participant's reaction to VR?

- Any suggestions or comments?

- Did they enjoy the games and asked the rationale behind them?

- Did they suggest other alternatives?

- Did participants provide information regarding previous experience with VR or video games? This could be relevant since it may affect awareness and interest, as well as acceptance

- Did participants want to repeat the games for longer periods?

Discussion

- Encephalograms are mentioned. But it would be relevant to have images to complement the text in the manuscript.

Conclusion

- Should be 'Conclusions and future Work'.

- I would like to have more discussion on the next steps of the author's work. 

Consider the following works:

- Andreikanich, A., Santos, B. S., Amorim, P., Zagalo, H., Marques, B., Margalho, P., ... & Dias, P. (2019). An exploratory study on the use of virtual reality in balance rehabilitation. In International Conference of the IEEE Engineering in Medicine and Biology Society (EMBC), pp. 3416-3419. 

- Paraense, H., Marques, B., Amorim, P., Dias, P., & Santos, B. S. (2022). Whac-A-Mole: Exploring virtual reality (VR) for upper-limb post-stroke physical rehabilitation based on participatory design and serious games. In IEEE Conference on Virtual Reality and 3D User Interfaces Abstracts and Workshops (VRW), pp. 716-717. 

Author Response

(The authors gave the same response as above.)

Round 2

Reviewer 1 Report

The authors of this study did a good job of responding to both reviewers. I am satisfied with the answers given and with the quality of the article presented.

Author Response

Dear Sir, dear Madam:

Thank you very much.
